# Prevention and Treatment of Chemotherapy and Radiotherapy Induced Oral Mucositis

**DOI:** 10.3390/medicina55020025

**Published:** 2019-01-22

**Authors:** Goda Daugėlaitė, Kristė Užkuraitytė, Eglė Jagelavičienė, Aleksas Filipauskas

**Affiliations:** 1Faculty of Medicine, Medical Academy, Lithuanian University of Health Sciences, A. Mickevičiaus 9, LT-44307 Kaunas, Lithuania; kriste.uzkuraityte@gmail.com; 2Department of Dental and Oral Pathology, Medical Academy, Lithuanian University of Health Sciences, Eiveniu 2, LT-50161 Kaunas, Lithuania; egle.jagelaviciene1@gmail.com; 3MediCA Klinika, V.Krėvės 53, LT-50358 Kaunas, Lithuania; aleksasfilipauskas@gmail.com

**Keywords:** oral mucositis, prevention, therapy, chemotherapy, radiotherapy

## Abstract

*Background and objectives*: Oral mucositis is one of the main adverse events of cancer treatment with chemotherapy or radiation therapy. It presents as erythema, atrophy or/and ulceration of oral mucosa. It occurs in almost all patients, who receive radiation therapy of the head and neck area and from 20% to 80% of patients who receive chemotherapy. There are few clinical trials in the literature proving any kind of treatment or prevention methods to be effective. Therefore, the aim of this study is to perform systematic review of literature and examine the most effective treatment and prevention methods for chemotherapy or/and radiotherapy induced oral mucositis. *Materials and methods*: Clinical human trials, published from 1 January 2007 to 31 December 2017 in English, were included in this systematic review of literature. Preferred reporting items for systematic reviews and meta-analysis (PRISMA) protocol was followed while planning, providing objectives, selecting studies and analyzing data for this systematic review. “MEDLINE” and “PubMed Central” databases were used to search eligible clinical trials. Clinical trials researching medication, oral hygiene, cryotherapy or laser therapy efficiency in treatment or/and prevention of oral mucositis were included in this systematic review. *Results*: Results of the studies used in this systematic review of literature showed that laser therapy, cryotherapy, professional oral hygiene, antimicrobial agents, Royal jelly, L. brevis lozenges, Zync supplementation and Benzydamine are the best treatment or/and prevention methods for oral mucositis. *Conclusions*: Palifermin, Chlorhexidine, Smecta, Actovegin, Kangfuxin, L. brevis lozenges, Royal jelly, Zync supplement, Benzydamine, cryotherapy, laser therapy and professional oral hygiene may be used in oral mucositis treatment and prevention.

## 1. Introduction

Oral mucositis (OM) refers to erythematous and painful ulcerative lesions of the oral mucosa observed in patients with cancer, who are treated with chemotherapy, and/or with radiation therapy [1]. According to the majority of studies, this complication occurs in up to 80% of patients receiving high-dose chemotherapy, and in up to 100% of patients receiving radiotherapy for head and neck cancer, and approximately 20–40% in those, who receive conventional chemotherapy [1,2].

OM is a painful complication that causes dysphagia, alterations in taste, weight loss, and secondary infections. These complications can significantly complicate treatment, extend hospitalization, and decrease the patient’s quality of life (QoL) [3]. The National Cancer Institute (NCI) published Common Terminology Criteria for Adverse Events (CTCAE). It includes separate subjective and objective scales for mucositis: Grade 1—Erythema of the mucosa; Grade 2—Patchy ulcerations or pseudomembranes; Grade 3—Confluent ulcerations or pseudomembranes; bleeding with minor trauma, Grade 4—Tissue necrosis; significant spontaneous bleeding; life-threatening consequences, Grade 5—Death [1]. It is important to keep in mind, however, that when chemotherapy or/and radiotherapy are applied, oral mucosal lesions are also possible in other erosive diseases, such as oral candidiasis [4], herpes simplex virus infection [4,5], acute Graft-versus-Host disease [6].

Nowadays, there are several models explaining the development of OM and its prevention and treatment strategies [7]. A five-stage chronological process explains the mechanism of pathogenesis: in the beginning, radiation and/or chemotherapy induce cellular damage and generation of free radicals resulting in death of the basal epithelial cells. It is followed by increase of inflammatory factors, which exaggerate cell death. Upregulation of pro-inflammatory cytokines cause mucosal ulcerations, which accelerate a secondary infection. In the last stage, epithelial proliferation as well as cellular and tissue differentiation occurs [2,8].

Up until now, there have not been any truly evidence-based clinical practice guidelines for treatment and/or prevention of OM [2]. Because there is not enough information about any OM treatment and prevention method effectiveness, it is necessary to research the latest clinical trials and summarize them. In 2014, The Multinational Association of Supportive Care in Cancer and International Society of Oral Oncology (MASCC/ISOO) published evidence-based clinical practice guidelines for mucositis. Treatment methods were categorized into seven groups: (1) basic oral care; (2) growth factors and cytokines; (3) anti-inflammatory agents; (4) anti-microbials, coating agents, anesthetics, and analgesics; (5) laser and other light therapy; (6) cryotherapy; and (7) natural and miscellaneous agents [1]. Therefore, the purpose of the study is to analyze the clinical trials which prove the effectiveness of pharmaceutical medications or treatment strategies in OM treatment and prevention, performed from 1 January 2007 until 31 December 2017.

## 2. Materials and Methods

The systematic review was conducted, publications were selected, and data was analyzed in accordance with the preferred reporting items for systematic reviews and meta-analysis (PRISMA) guidelines [9]. Bibliographic searches were carried out in PubMed for recent clinical trial studies published between 1 January 2007 and 31 December 2017. The analyzed studies were named as clinical randomized trials, quoted and published in journals in English. Only studies with humans and at least 100 participants were included. The results of the studies evaluated at least one of the following treatment or prevention outcomes: OM degree at the end of the trial according to the NCI-CTCAE classification, OM development or remission, duration of manifestation, time of occurrence, oral mucosa or QoL according to subjective patient complaints, need of opioids because of OM. This systematic review of the literature did not include certain articles, the results of which discuss the occurrence of treatment complications, concomitant diseases, infections and gastrointestinal mucositis.

### 2.1. Data Extraction

Three reviewers independently assessed the titles and abstracts of articles to determine the trial inclusion. Information was extracted from the full texts, using a predefined data extraction sheet. Disagreements were resolved by discussion. In the search for publications, keyword combinations were used: “oral mucositis” OR “oral mucositis treatment” OR “oral mucositis prevention” OR “oral mucositis therapy” OR “chemotherapy adverse events” OR “radiotherapy adverse events” OR “oral mucositis classification” NOT “graft versus host disease”.

Articles were included if they matched the following selection criteria, according to the characteristics of the study:The sample was at least 100 subjects, older than 18 years old, who have not been diagnosed with OM, but who have been receiving or have been planning to undergo chemotherapy or radiotherapy.Clinical randomized trials that were investigated:2.1The efficacy of medicine in the treatment or prevention of OM.2.2The effectiveness of the chosen treatment method in the prevention or treatment of OM.The analyzed factors were at least one of the following treatment or prevention outcomes:3.1OM degree at the end of the study according to the NCI-CTCAE classification;3.2OM development or remission;3.3OM incidence duration factor;3.4The time of occurrence of OM;3.5Condition of OM or QoL evaluation according to subjective patient opinion;3.6Need for opioids because of OM.The research provides statistical analysis of the data by comparing groups with different treatments or prevention methods.

### 2.2. Data Synthesis

A total of 32,483 articles were identified during the search of articles. After activating the filters, 5455 of the articles were selected, and after reviewing the summaries—55. The text of all studies was read to the fullest extent and, following the application of the selection criteria, 21 articles were left to the final analysis (the complete procedure for the selection of the literature is shown in Figure 1). 3230 people aged over 18 participated in the studies. Their average age ranged from 47 to 62 years. All samples were treated with any of oncologic treatment methods.

A total of 20 out to 21 studies indicated localization of an oncological disease: hematologic cancer [2], colon cancer [3], head and neck cancer [10,11], hematologic cancer [12,13,14], head and neck cancer [15,16,17,18,19,20], hematologic cancer [21], breast cancer [22], head and neck cancer [23,24,25,26,27], hematological cancer [28]. In all studies, the oncologic treatment method is described: chemotherapy [2], combined chemotherapy and radiotherapy [3,10,11], radiotherapy [11], chemotherapy [12,13,14], combined chemotherapy and radiotherapy [15,16], radiotherapy [17,18], combined chemotherapy and radiotherapy [18], chemotherapy [19], combined chemotherapy and radiotherapy [20], radiotherapy [20], chemotherapy [21], combined chemotherapy and radiotherapy [22,23], radiotherapy [24], combined chemotherapy and radiotherapy [25], radiotherapy [26], combined chemotherapy and radiotherapy [27], chemotherapy [28].

## 3. Results

In total, the study analyzed 21 articles in which one or more methods of treatment or efficacy of any drug for the treatment or/and prophylaxis of OM were studied. In general, 18 studies chose the OM degree to estimate the final treatment outcomes [2,10,11,12,13,14,15,17,18,19,20,21,22,23,24,25,27,28]. 2 studies chose the manifestation of progression or remission of this disease [13,22], 7 studies—the duration of the disease [3,15,19,20,21,24,26] and 3 attempted to determine the time of occurrence of OM after the oncologic treatment had started [23,26,27]. One study evaluated the results according to questionnaires in which patients described their oral mucosa condition subjectively [16]. In addition, in two studies, the authors presented the results of treatment based on the need for opioids [12,25]. Systematized results are presented in Table 1.

### 3.1. Review of Studies’ Results

#### Professional and Individual Oral Hygiene

In a study carried out by Kashiwazaki et al. [2], the effect was investigated to assess professional oral hygiene (POH) on the prevention of OM in pre and post bone marrow transplantation patients receiving a high dose chemotherapy course. OM degree had been evaluated on a daily basis. The results showed that patients, receiving POH, had a statistically significantly lower possibility to get OM, than patients with no POH (*p* < 0.001).

Yokota et al. [10] investigated the effect of individual oral hygiene (IOH) on OM severity. In total, 120 patients receiving chemoradiotherapy treatment participated in the study. Patients were instructed about IOH. OM degrees were evaluated based on clinical examination results and subjective complaints from patients. The results had shown that, according to clinical outcomes, 42.5% of patients developed grade III or IV of OM, and 53.3%—according to subjective complaints. The conclusion of the study suggested that the IOH was not an effective method for the reduction of OM severity.

### 3.2. Medications

#### 3.2.1. Growth Factors and Cytokines

Wu et al. [11] studied the effect of recombinant human epidermal growth factor (RhEGF) in reducing the degree of OM. The study involved 113 patients receiving chemoradiotherapy. Patients were assigned to a placebo group (n = 28) or to 1 of 3 EGF-treatment groups (10 (n = 29), 50 (n = 29) or 100 (n = 27) μg/mL doses, delivered in a spray, twice daily). If the degree of OM was ≤ II, RhEGF was considered effective in prevention and treatment. RhEGF significantly reduced the incidence of severe OM at the primary endpoint (a 64% response was observed with 50 μg/mL EGF vs. a 37% response in the control group; *p* = 0.0246). Kim et al. conducted a similar study. [12]. 138 patients were divided into 2 groups: control—placebo group and RhEGF treatment group (50 μg/mL doses, twice daily). Treatment was considered effective if the degree of OM was ≤ I. The results of this study showed no statistically significant difference between these groups (*p* = 0.717). So, according to both studies, controversial results were obtained, therefore, further and more detailed studies are needed.

Bradstock et al. [13] studied the effect of Palifermin (keratinocyte growth factor (KGF)), given 60 μg/kg daily IV for 3 days before and after chemotherapy, for mucosal protection. 155 subjects, who received combined chemotherapy, were included in the study (76 to palifermin and 79 to placebo groups). The results had shown that the severity of OM was reduced more significantly in the Palifermin group than in the placebo group (*p* = 0.007).

Blijlevens et al. [14] performed a study with the same medication, but only in patients, who received high-dose chemotherapy (n = 277). Patients were divided into 3 groups: group I—placebo (n = 57), group 2 received six doses of Palifermin before and after chemotherapy (n = 113) and group 3 received three doses of Palifermin only before chemotherapy (n = 107). The results showed no statistically significant differences between grades III (*p* = 0.25) or IV (*p* = 0.66) of OM and Palifermin usage before and after chemotherapy or just before chemotherapy, compared to placebo group. Severe OM occurred in 37% (placebo), 38% (pre-/post-chemotherapy) and 24% (pre-chemotherapy) patients.

Quynh-Thu Le et al., also studied the effect of Palifermin on OM prevention. [15]. The study involved 188 patients treated with chemoradiotherapy. 94 subjects received placebo treatment and 94—assigned to the study group receiving Palifermin at 180 μg/kg. The results had shown that grade III or grade IV of OM in the Palifermin group was statistically significantly lower than in the placebo group (*p* = 0.041).

Hoffman et al. [16] investigated the effect of granulocyte macrophage-colony stimulating factor (GM-CSF) in the prevention and treatment of radiotherapy-induced OM. 58 subjects were treated with GM-CSF and 56 received placebo. The GM-CSF daily dose was 250 μg/m2 for a week before and it was stopped two weeks after radiation completion. The respondents assessed their condition of the oral mucosa and the QoL before and after treatment. The results showed that there was no statistically significant difference of QoL in total symptom score between both groups (*p* > 0.05). However, patients receiving GM-CSF reported higher amount of mucous (*p* = 0.008) than placebo patients.

#### 3.2.2. Anti-Inflammatory Medications

Kazemian et al. [17] evaluated the effect of benzydamine oral rinse (non-steroidal anti-inflammatory drug (NSAID) for prevention of radiation-induced mucositis. The study involved 100 patients divided into benzydamine and placebo groups. The results showed that in the benzydamine group, the frequency of mucositis grade ≥III was 43.6%, in contrast to 78.6% in the placebo group (*p* = 0.001). Grade III mucositis was 2.6 times more frequent in the placebo group (*p* = 0.049).

Rastogi et al. [18] also studied the effect of this medication on the prevention of radiotherapy and chemotherapy-induced OM. The study involved 120 respondents. The results showed that patients receiving radiotherapy and benzydamine oral rinse had grade III of OM statistically significantly less often than the control group (*p* = 0.038). However, no statistically significant data were obtained from the chemotherapy-treated patients (*p* = 0.091).

#### 3.2.3. Antimicrobial Medication

Sorensen et al. [19] studied the effect of chlorhexidine mouth rinse for the treatment and prevention of chemotherapy induced OM. The study involved 206 subjects, who were divided into 3 groups: chloroxidine mouth rinse group, placebo group (treated using normal saline) and cryotherapy group. The results showed that in the chlorhexidine rinse group, OM grade III or IV was statistically significantly less frequent than in the group receiving normal saline (*p* < 0.01). In addition, OM duration was statistically significantly longer in the group that used normal saline (*p* = 0.035).

Wong et al. [20] studied the effect of antibacterial rinse Caphosol^®^ mouthwash (EUSA Pharma, Dublin, Ireland) on the treatment and prevention of radiotherapy induced OM. Respondents were divided into two groups: intervention (n = 108) and control (n = 107). The results showed that OM grade IV was less likely to develop in the group of Caphosol mouthwash users, but the results were not statistically significant (*p* = 0.839); additionally, there was no statistically significant difference of manifestation time of OM in both groups (*p* = 0.692).

Lin et al. [21] compared the efficiency of dioctahedral smectite and iodine glycerin (DSIG) cream for prevention and treatment of chemotherapy induced OM. The study sample consisted of 130 subjects that were divided into 2 groups: one group was treated with DSIG cream (n = 63) and the other received placebo mouthwash treatment (n = 67). The results showed that the group, treated with DSIG cream had statistically significantly shorter time of OM incidence than the placebo group (*p* < 0.001). The group treated with DSIG cream had a statistically significantly lower degree OM than the group treated with placebo (*p* < 0.001).

Wu et al. [22] studied the effect of Actovegin in the treatment and prevention of chemoradiotherapy induced OM. The study involved 156 patients that were divided into 3 groups: the effectiveness of prevention in patients taking Actovegin from the first day of chemoradiotherapy was studied in the first group (n = 53). The effectiveness of treatment, when patients started using Actovegin on the occurrence of grade II OM was studied in second group (n = 51). The third group did not receive any treatment (n = 52). The results showed that grade III of OM occurred statistically significantly less often in the first than in the third group (*p* = 0.002). However, there was no statistically significant difference in OM reduction between group 2 and 3 (*p* = 0.093). It was also noticed that in both the first (*p* = 0.023) and the second group (*p* = 0.035), OM was less likely to progress from grade II to grade III, than in the non-treated group.

Luo et al. [23] studied the effect of an antimicrobial medication Kangfuxin Solution, a pure Chinese herbal medicine, on the treatment of chemoradiotherapy induced OM. 215 patients were divided into 2 groups: the first group (n = 107) received Kangfuxin 3 times a day during the entire chemoradiotherapy or until grade 3 OM occurred, the second group (n = 108), was a control group and was given the same amount of borax gargle. The results showed that in group 1, OM of any degree developed statistically significantly less frequently than in the control group (*p* = 0.0084). In addition, the results showed that I° (*p* < 0.0001), II° (*p* = 0.0014) and III° (*p* = 0.0001) of OM occurred statistically significantly later than in the control group.

#### 3.2.4. Natural Medication

Bardy at al. [24] studied the effect of active Manuka honey in prevention of radiation-induced OM. The research sample consisted of 131 patients divided into 2 groups: first group received Manuka honey and the placebo group received sugar syrup. The results of the research were not statistically significant.

Erdem at al. [3] studied the effect of royal jelly in oral mucositis in patients undergoing radiotherapy and chemotherapy. The study population consisted of 103 patients who were divided into 2 groups: the royal jelly consuming group (n = 51) and the control placebo group (n = 52). All patients received mouthwash therapy with benzydamine hydrochloride and nystatin rinses. The results showed that the patients in the first group recovered statistically significantly faster: grade III OM healed in 3.5 days on average. Control group healed in 10 days on average (*p* = 0.005); grade II OM healed in 3 days, control group healed in 5.8 days (*p* = 0.0001); grade I OM healed in 1.1 day and control group healed in 2.7 days (*p* = 0.0001).

Sharma at al. [25] studied the effect of administering Lactobacillus brevis CD2 lozenges on the incidence and severity of mucositis and tolerance to chemo-radiotherapy. The study treatment was given during and for 1 week after completion of anticancer therapy. The study involved 188 subjects divided into 2 groups: the group receiving a study treatment (n = 93) and a placebo group (n = 95). The results showed that OM of any degree developed statistically significantly less frequently in the L. brevis CD2 arm (*p* < 0.001). It also showed that patients receiving study treatment were statistically significantly less likely to use analgesics to relieve pain caused by OM (*p* = 0.02).

Lin at al. [26] studied the effect of zinc supplementation on the prevention of radiation-induced mucositis. The research involved 100 subjects divided into 2 groups of 50 subjects: group that received zinc supplementation and the placebo group. The results showed that in the group receiving zinc supplementation, grade II (*p* = 0.009) or grade III (*p* = 0.001) OM occurred statistically significantly later than in the placebo group. It was also observed that, in the group receiving zinc supplementation, ≥II grade OM lasted statistically significantly shorter than in the placebo group (*p* = 0.033).

### 3.3. Laser Therapy

Gautam at al. [27] studied the effect of low intensity laser therapy (LLLT) for the prevention and treatment of concurrent chemoradiotherapy induced OM. Research involved 221 subjects, who were divided into two groups: first group received LLLT 5 times per week for 6 anatomical oral cavity areas (n = 111), and a placebo group (n = 110). Both groups received treatment throughout the course of chemoradiotherapy. The results showed that in the first group, OM of any degree developed statistically significantly later than in the control group (*p* < 0.0001), also grade IV OM occurred statistically significantly less often than in the control group (*p* < 0.0001).

### 3.4. Cryotherapy

In the previously mentioned study of Sorensen at al. [19], the effect of cryotherapy on OM prevention was also examined. The test group used a crushed ice for 45 min during chemotherapy. The results showed that the grade III or IV OM developed statistically significantly less frequently in the group of cryotherapy patients, than in the control group (*p* < 0.005). Moreover, the duration of the disease was statistically significantly longer in the control group than in the cryotherapy group (*p* = 0.003).

Vokurka at al. [28] also analyzed the effects of cryotherapy. 126 patients receiving high-dose chemotherapy were included in the study. They were divided into 2 groups: a group of patients treated by cryotherapy (n = 36) and a control group (n = 90). Cryotherapy was performed by holding a piece of ice in the mouth during the chemotherapy infusion. The results showed that OM of any degree developed statistically significantly less frequently in the cryotherapy group (*p* ≤ 0.0001).

## 4. Discussion

OM is one of the most common complications during chemotherapy and radiotherapy. Although the information found in the literature so far suggests that currently there is no fully effective method for treating or preventing OM, on the basis of the studies analyzed in this scientific literature review, it can be said that it is possible to reduce clinical manifestations of this disease, or at least prevent, a more severe degree. However, it should be stressed that controversial results have been obtained in studies that have examined the same treatment or prevention method.

Medications of the growth factor subgroup showed different results: The efficacy of RhEGF in OM treatment was being analyzed in two studies, but results were positive only in a study conducted by Wu and co-authors [11]. In the study of Kim with co-authors [12], the efficacy of RhEGF was not proven. Controversial results were obtained possibly because the first [12] study was considered successful if it did not develop into II, III or IV degree of OM, and the next study was considered successful if it [11] did not develop only into III or IV degree of OM. While studying the effects of GM-CSF [16], no positive effect of the drug was observed, but it should be noted that oral mucosal condition was assessed in this study only by subjective patient complaints. Another drug, Palifermin, has been examined in three studies [13,14,15], but only one study [15] produced statistically significant results. However, the overall doses of Palifermin used in studies [13,14] were lower, and this could have affected the results of the research. Therefore, we cannot compare the products belonging to this subgroup to one another, because of the different criteria for the evaluation of the results.

In the anti-inflammatory drug subgroup, only benzidamine hydrochloride was examined. This product was effective in both OM treatment and prevention, but positive results were only obtained in patients receiving radiotherapy [17,18]. Such treatment results may be due to the different effects of chemotherapy and/or radiation therapy on the oral mucosal tissues.

In the antimicrobial product subgroup, significant therapeutic and preventive effects were obtained with the use of chlorhexidine [19], Smecta [21], Actovegin [22] and Kangfuxin [23], but comparing the efficacy of these products would be difficult due to the varying assessment criteria and treatment protocols chosen by researchers.

In the subgroup of natural products, significant therapeutic and preventive effects were found in royal jelly [24], *Lactobacillus brevis* lotion [25] and zinc supplementation [26]. It would be difficult to compare the efficacy of these products with each other, as the authors have chosen different criteria for the evaluation of results in all studies.

Low intensity laser therapy and cryotherapy were attributed to the subgroup of physiotherapy. Two cryotherapy studies have been found [19,28] with positive treatment and prevention results. Such results were likely because the cold prevents blood flow to the oral cavity, resulting in reduced cytotoxic chemotherapy access to the mucosal tissues, which reduces the likelihood of OM, but also reduces the effectiveness of primary disease treatment. Therefore, the use of cryotherapy is debatable. In addition, the efficiency of low intensity laser is debatable. Although the results were positive, only one study was found. Therefore, so as to state that this method is truly effective, further research is needed.

This systematic analysis has shown that in the studies, different protocols for the evaluation of the results were used to evaluate the efficacy of the treatment method or the efficacy of the medication: it was based on different indicators, different therapeutic doses were applied, and professional competencies of assessors differed. Some supplements or treatments have been evaluated only in one or two studies, so the validity of their efficacy is debatable, and similar ongoing studies are needed. To achieve statistically significant results, the review selected studies with a sample size greater than 100. Therefore, it can be said that a number of studies with the potential for OM treatment or prevention have not been included in this work. Thus, in the case of all of the above-mentioned shortcomings, it is difficult to provide generalized clinical recommendations.

## 5. Conclusions

This systemic review of data from 21 CRCT provided evidence that IOH does not help to prevent OM, but POH does; therefore, it is an important preventive and therapeutic tool. Although there are not much data, but there are suggestions based on the studies, which indicate that medicines like Palifermin, chlorhexidine, Smecta, Actovegin, Kangfuxin, L. Brevis lotion, royal jelly and zinc supplement are effective medicines and can be used to treat and prevent chemoradiotherapy induced OM. Benzydamine is also effective, but only after radiotherapy. The efficacy of RhEGF has not been demonstrated, and GM-CSF and Caphosol had no effect on the healing of OM, so they are not applicable in prophylaxis. Considering the impact of physiotherapy for OM treatment and prevention, low-intensity laser therapy and cryotherapy reduce the development and duration of OM. However, additional long-term research is needed to develop precise guidelines for the treatment and prevention of chemoradiotherapy induced OM.

## Figures and Tables

**Figure 1 medicina-55-00025-f001:**
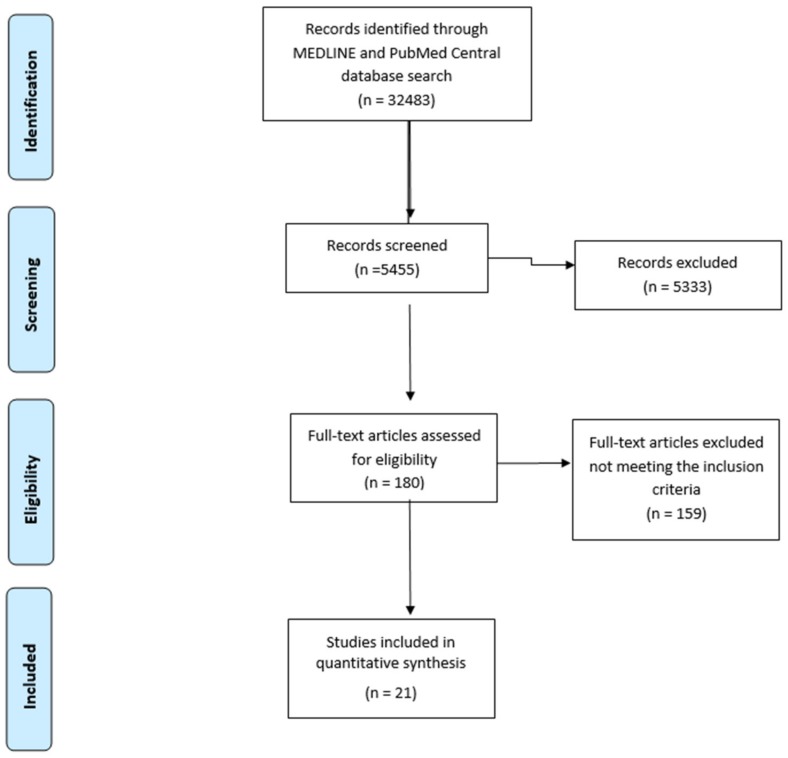
Schematic diagram explaining the assortment of studies/reports (2009 preferred reporting items for systematic reviews and meta-analysis (PRISMA) flow diagram).

**Table 1 medicina-55-00025-t001:** Study characteristics.

No.	First Author Year	Sample Size (n)	Design	Treatment Method	Selected Evaluation Criteria	Main Results
1.	Kashiwazaki, H. 2012	140	CRCT	POH	≥I° OM devel.	Patients, who had received POH have a lower probability of developing OM in any degree than patients with no POH (*p* < 0.001).
2.	Yokota, T. 2016	120	CRCT	IOH	≥II° OM devel.	The results showed that this treatment method was ineffective.
3.	Wu, H.G. 2009	113	CRCT	RhEGF	≥II° OM devel.	In the group receiving the RhEGF 50 μg/mL dose, the ≤ II° OM developed less frequently (*p* < 0.05).
4.	Kim, J.W. 2017	138	CRCT	RhEGF	≤I° OM devel., duration of opioid use	There is no statistically significant difference between the control group and the treatment group in OM devel. (*p* = 0.717). In the RhEGF group, opioid use was shorter (*p* = 0.036).
5.	Bradstock, K.F. 2014	155	CRCT	Palifermin	≥III° OM devel., decrease of OM degree	There was no significant association between the ≥III° OM in the Palifermin-treated and control group (*p* = 0.21). In the group receiving Palifermin, the severity of the disease decreased more than in the control group (*p* = 0.007).
6.	Blijlevens, N. 2013	277	CRCT	Palifermin	≥III° OM devel.	The use of Palifermin before and after chemotherapy did not decrease the III° (*p* = 0.25) or IV° (*p* = 0.66) of OM, also was only used before chemotherapy—III° (*p* = 0.81) or IV° (*p* = 0.81).
7.	Le, Q.T. 2011	162	CRCT	Palifermin	≥III° OM devel., IV° OM devel. duration	There was a lower incidence of ≥III° OM in the Palifermin group than in the placebo group (*p* = 0.041). A shorter duration of time of severe OM degree was observed in the Palifermin group than in the placebo group (*p* = 0.016).
8.	Hoffman, K. 2014	114	CRCT	GM-CSF	QoL, subjective estimation of oral mucosa condition	There was no statistically significant difference of QoL in total symptom score between both groups (*p* > 0.05). Patients receiving GM-CSF reported higher amount of healthy mucous (*p* = 0.008) than placebo patients.
9.	Kazemian, A. 2008	100	CRCT	Benzydamine	≥III° OM devel.	In the group treated with benzydamine, ≥III° OM devel. less often than in the placebo group (*p* = 0.049).
10.	Rastogi, M. 2017	120	CRCT	Benzydamine	III° OM devel.	For patients, who used benzydamine and received radiotherapy, III° OM occurred less often than in control group (*p* = 0.038). No statistically significant differences were found in patients who were receiving chemotherapy (*p* = 0.091).
11.	Sorensen, J.B. 2008	206	CRCT	chlorhexidines, cryotherapy.	≥III° OM devel., ≥III° OM devel. duration	In the chlorhexidine rinse group, OM III° or IV° was statistically significantly less frequent (*p* < 0.01). OM continued to stay longer in the group that used normal saline (*p* = 0.035). The III° or IV° OM developed less frequently in the group of cryotherapy patients than in the control group (*p* < 0.005), also duration of the disease was also longer in the placebo group (*p* = 0.003).
12.	Wong, K.H. 2017	215	CRCT	Caphosol	IV° OM devel., IV° OM devel. duration	OM grade IV was less frequently developed in the group of Caphosol mouthwash users, but the results were not statistically significant (*p* = 0.839), also there was no statistically significant difference of OM manifestation time in both groups (*p* = 0.692).
13.	Lin, J.X. 2015	130	CRCT	DSIG	≥I° OM devel., OM devel. duration	The group treated with DSIG cream had shorter time of OM incidence (*p* < 0.001) and lower degree (*p* < 0.001) OM than placebo group patients.
14.	Wu, S.X. 2010	156	CRCT	Actovegin	III° OM devel., OM degree progression	III° OM occurred less in the prevention group than in the group without treatment (*p* = 0.002). There was no statistically significant difference between groups in OM reduction (*p* = 0.093). In both—the group received preventive treatment (*p* = 0.023) and in the group that received treatment only after symptoms occurred (*p* = 0.035), OM was less likely to progress from II° to III° than in the non-treated group.
15.	Luo, Y. 2016	215	CRCT	Kangfuxin	≥I° OM devel., I°, II°, III° OM devel. time	In the group that received Kangfuxin, OM of any degree developed less frequently than in the control group (*p* = 0.0084). Also, I° (*p* < 0.0001), II° (*p* = 0.0014) and III° (*p* = 0.0001) OM occurred later than in the control group.
16.	Gautam, A.P. 2012	221	CRCT	LLLT	IV° OM devel., ≥I° OM devel. time	In the group that received LLLT, OM of any degree developed later and IV° OM was less common than in control group (*p* < 0.0001).
17.	Vokurka, S. 2011	126	CRCT	Cryotherapy	≥I° OM devel., ≥III° OM devel.	OM of any degree developed less frequently in the cryotherapy group (*p* ≤ 0.0001).
18.	Bardy, J. 2012	131	CRCT	Manuka Honey	III OM devel.; ≥I° OM devel. duration	There was no statistically significant difference in the devel of III° OM (*p* = 0.64) and duration of occurrence (*p* = 0.79) between the Manuka honey and placebo groups.
19.	Erdem, O. 2014	103	CRCT	Royal jelly	I°, II°, III° OM recovery duration	Patients in the group that received royal jelly recovered healed faster: III° OM recovered in 3.5 days on average, control group healed recovered in 10 days on average (*p* = 0.005); II° OM recovered in 3 days, control group recovered in 5.8 days (*p* = 0.0001); I° OM recovered in 1.1 day and control group recovered in 2.7 days (*p* = 0.0001).
20.	Sharma, A. 2012	188	CRCT	*L. brevis* lozenges	≥I° OM devel., analgesic necessity	OM of any degree developed less frequently in the *L. brevis* CD2 arm (*p* < 0.001) and these patients were less likely to use analgesics to relieve pain caused by OM (*p* = 0.02).
21.	Lin, Y.S. 2010	100	CRCT	Zinc supplementation	II°/III° OM devel. time, ≥II° OM devel. duration	In the group that received Zinc supplementation, II° (*p* = 0.009) or III° (*p* = 0.001) OM occurred later than in the placebo group. In the group receiving zinc supplementation, ≥II° OM lasted shorter than in the placebo group (*p* = 0.033).

n = sample size, CRCT = clinical randomized controlled trial, POH = professional oral hygiene, IOH = individual oral hygiene, devel. = development, DSIG = Dioctahedral smectite and iodine glycerin, LLLT = low level laser therapy.

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
