# Peer review of "Prevention and Treatment of Chemotherapy and Radiotherapy Induced Oral Mucositis"

_medicina, 2019, doi:10.3390/medicina55020025_

Round 1
Reviewer 1 Report
The purpose of the study is to analyze clinical trials proving pharmaceutical medications or treatment strategies effectiveness in OM treatment and prevention performed from 1/1/2007 till 63 31/12/2017.
The work is well structured and designed is very interesting however some gaps should be considered
databases were conducted Searches of Medline, onlay Embase, Scopus. ????
It would have been interesting to indicate the treatment of mucosistis induced by each of the drugs and type of tumor AND / or by radiotherapy dose
Author Response
Thank you very much for your comments. We are very pleased that you are interested in our manuscript.
Point 1: databases were conducted Searches of Medline, onlay Embase, Scopus. ????Response 1: Databases were only conducted by searches of Medline, because we decided that it is the most accessible and most publishing database. But we are thankful for your comment, we will definitely use your advice in other studies.
Point 2: it would have been interesting to indicate the treatment of mucosistis induced by each of the drugs and type of tumor AND / or by radiotherapy dose .
Response 2: we were thinking about it even before preparing this study, but all authors decided, that it might make the text difficult ant too detailed/complex, cause most of the studies did not indicate the drugs or radiotherapy doses. So, according to Your comment, we have added a paragraph specifying what exact treatment method was used and what type of cancer was diagnozed for the respondents in all studies (page 3, lines 113-117) (Text: 20 out of 21 studies indicated localization of an oncological disease: head and neck cancer [10,11,15,16,17,18,19,20,23,24,25,26,27]; hematologic cancer [2,12,13,14,21,28]; breast cancer [22]; colon cancer [3]. In all studies, the oncologic treatment method is described: chemotherapy [2,12,13,14,19,21,28]; radiotherapy [11,17,18,20,24,26]; combined chemotherapy and radiotherapy [3,10,11,15,16,18,20,22,23,25,27].

Reviewer 2 Report
Dear Editor, Thank you for giving me the opportunity to review this systematic review of the literature on therapeutic strategies in oral mucositis. The topic is of great interest both for the scientific community and for clinical practice. However, some considerations must be made:
1- The registration to an international database like "PROSPERO" is missing, which ascertains the correct revision methodology as well as originality.
2- Why have you chosen a sample size so high in materials and methods (100)? There are extremely interesting studies even with reduced sample sizes. For example, there is a very interesting study that could give more data on palifermin (Lauritano D et al.) Clinical effectiveness of palifermin in prevention and treatment of oral mucositis in children with acute lymphoblastic leukemia: a case-control study. 2014 Mar; 6 (1): 27-30)
3- Why was not a meta-analysis done, given the abundance of available data, perhaps narrowing out the outcome?
4- In the introduction we should mention the fact that oral mucositis is a very precise pathological picture and the patient's clinical history and oral localization allow us to make differential diagnosis with other erosive diseases such as OLP (I suggest Lucchese A, Dolci A, Minervini G, Salerno C, DI Stasio D, Minervini G, Laino
L, Silvestre F, Serpico R. Vulvovaginal gingival lichen planus: report of two cases and review of literature. Oral Implantol (Rome). 2016 Nov 13; 9 (2): 54-60.
Finally, English seems to be a little elementary with some syntactical errors and should be revised.
I suggest publication only after major revisions.
Author Response
Thank You very much for Your comments. We are very pleased that You are interested in our manuscript. And we hope, we've made the manuscript corrections properly.
Point 1: The registration to an international database like "PROSPERO" is missing, which ascertains the correct revision methodology as well as originality.
Response 1: Thank You for Your note, but during the christmas and new year period PROSPERO is not working and we can't use it. Therefore, we do not know whether to wait for this site to work again or to send it to another website for verification. Maybe You could offer another option? However, we can ensure that the manuscript is original, authors collected the information and made the systematic review by themselves.
Point 2: Why have you chosen a sample size so high in materials and methods (100)? There are extremely interesting studies even with reduced sample sizes. For example, there is a very interesting study that could give more data on palifermin (Lauritano D et al.) Clinical effectiveness of palifermin in prevention and treatment of oral mucositis in children with acute lymphoblastic leukemia: a case-control study. 2014 Mar; 6 (1): 27-30).
Response 2: Studies with smaller samples were also very interesting, as did the example of the study You sent. But we decided to choose a sample size so high in materials and methods (100) because we hoped that the larger the sample we choose, the more statisically significantly the results will be, and the more reliable the systematic review will be.
We will definitely use Your advice in developing a new study. Maybe we will try to perform a systematic review about each treatment or prevention method of oral mucositis, by reviewing all studies, regardless of sample size.
Point 3: Why was not a meta-analysis done, given the abundance of available data, perhaps narrowing out the outcome?
Response 3: Thank You for Your offer, but this time we decided to use systematic review method. Next time we will take Your advice and will do meta-analysis.
Point 4: In the introduction we should mention the fact that oral mucositis is a very precise pathological picture and the patient's clinical history and oral localization allow us to make differential diagnosis with other erosive diseases such as OLP (I suggest Lucchese A, Dolci A, Minervini G, Salerno C, DI Stasio D, Minervini G, Laino L, Silvestre F, Serpico R. Vulvovaginal gingival lichen planus: report of two cases and review of literature. Oral Implantol (Rome). 2016 Nov 13; 9 (2): 54-60.
Response 4: we used Your advice and included it in the manuscript. Correction was made on page number 2 (lines 47 - 50). However, after discussing with the authors, we decided that a more detailed description of other diseases similar to oral mucositis is not required, so we mentioned about differential diagnosis and we have identified several sources in which the reader could easily find reliable information about these diseases.
Point 5: Finally, English seems to be a little elementary with some syntactical errors and should be revised.
Response 5: We are very sorry for the lack of professionalism at this point. Manuscript was checked by professionals and all corrections are noted in the text.

Reviewer 3 Report
The article by Daugelaite et. al is a nice summary of prior studies on the treatment option for oral mucositis caused by radiotherapy and chemotherapy. As the authors suggested radiotherapy always resulted in oral mucositis while it’s also very prevalent in chemotherapy-treated patients. At present, there is not a firm guidelines by any organization which suggest effective treatment of oral mucositis. However, it badly effect the quality of life of patients. Therefore, this area is worth investigating. Authors have done a good job of compiling prior works on the treatment of oral mucositis. Although studies are very few but there are suggestions based on those studies which suggest that medicine like Palifermin, chlorhexidine, Smecta, Actovegin, Kangfuxin, L. brevis lotion, royal jelly, and zinc supplement; and low-intensity laser and Cryotherapy are the very important treatment for oral mucositis. This review brings the point to the table of scientist that in which direction they need to give more time in research related to oral mucositis. I guess this topic will also raise awareness in the readers of this journal. I find this article appropriate for this journal.
Author Response
Thank You very much for Your comments. We are very pleased that You are interested in our manuscript. It is a real honor for us. Again, great thanks to You.
Reviewer 4 Report
The authors reviwed and analyzed clinical trials proving pharmaceutical medications or 62
treatment strategies effectiveness in oral mucotitis treatment and prevention.
For preventing severe mucotitis induced by CRT, the authors should add discussion in terms of supportive therapy such as nutrition using PEG tube or NG tube.
Author Response
Thank You very much for Your comments.
Point 1: For preventing severe mucotitis induced by CRT, the authors should add discussion in terms of supportive therapy such as nutrition using PEG tube or NG tube.
Response 1: We are very thankful for your advice, but our systematic review was made based on some criterias (page 3, lines 89-105) and supportive therapy is not one of them, that is why we do not discuss it in our review.
We understant that supportive therapy topic is very important for patients' with OM quality of life, so in our opinion in this case another study might be needed.
So we hope, that You will approve our option.

Round 2
Reviewer 2 Report
Dear author,
Thank you for making the changes I requested. Next time before setting up a systematic review, I'm sure this will first be registered on PROSPERO. As for the other required corrections, they were made, and therefore the article can be considered for publication as it is.
Best regards
Reviewer 4 Report
well presented review